# Numerical Simulation of Co-Barrier Efficiency of Air Curtains in Mine Refuge Chambers

Zhenqiang Shen [1], Zujing Zhang [1,2,*], Jiang Lan [1], Xusong Tian [1], Hong Ma [1] and Ruiyong Mao [1,2,*]

1    College of Civil Engineering, Guizhou University, Guiyang 550025, China
2    Guizhou Provincial Key Laboratory of Rock and Soil Mechanics and Engineering Safety, Guiyang 550025, China
*    Correspondence: zjzhang3@gzu.edu.cn (Z.Z.); rymao@gzu.edu.cn (R.M.); Tel.: +86-185-2391-9513 (Z.Z.); +86-139-8505-6628 (R.M.)

**Abstract:** Toxic and harmful gases may follow evacuees into the mine refuge chamber, endangering their life during the mine disaster. Gases from the outside environment are successfully kept out of the mine refuge chamber by air curtains. The effect of the air curtain installation position, jet velocity, angle, and structural parameters on the CO barrier efficiency is studied using numerical simulation of computational fluid dynamics (CFD) in this paper. The results are also used to show the influence of the above factors on the efficiency of the air curtain. The results show that: (1) increasing the air curtain jet speed does not always increase or even decrease the air curtain barrier efficiency. (2) The best CO barrier efficiency of 68.1% was obtained by an air curtain installed on the top side of the refuge door with a velocity of 22 m/s and a jet angle of $10°$. (3) A mathematical relationship between air curtain structural parameters and barrier efficiency is developed to provide a reference for the design of air curtain pipeline structures.

**Keywords:** air curtain; mine refuge chamber; barrier efficiency; structural parameters; mathematical relationship

## 1. Introduction

The underground mine is a dangerous production site due to the accidents of gas explosions, coal dust, fire, and roof hazards that can occur during the mining of coal [1]. Statistics and analysis of the majority of coal mine accidents show that most miners die from drowning, asphyxiation, or CO poisoning due to their inability to escape from the mine after water penetration or fires, or explosions, or to escape from the high temperatures, toxic and harmful gas accident site [2]. More than 80% of victims of coal mine gas explosions and fire accidents died from CO poisoning and hypoxia asphyxia [3–5]. Mine refuge chambers are regarded to be the most important life-saving equipment, providing evacuees with secure, airtight locations to survive mine accidents by removing them from the harmful environment, and providing for their basic needs for at least 96 h [6]. During disasters in mines, evacuees open the refuge anti-blast door first and then enter the refuge from the outside environment. CO gases can also enter the refuge and endanger the lives of the evacuees [7].

Air curtains can isolate protected areas from toxic and hazardous environments by generating one or more high-motion parallel air flows [8]. Installing air curtains in front of the refuge door can prevent harmful gases from entering the refuge [9]. Based on such a fact, air curtains can reduce the infiltration of heat, gas, and mass from one space to another without occupying space and without affecting the personnel passage, it has been widely applied and studied in the fields of dust control [10,11], fire prevention in building [12–14], and smoke prevention in refuge [15].

Experimenting is the best method for evaluating how air curtains perform [16]. However, this way is inefficient in terms of cost and time [17]. CFD numerical simulation is

used widely to study air curtains because of its high accuracy, easy operation, and low cost [18,19]. Krajewski and Węgrzyński [20] found by bench experiments and FLUENT that air curtains can effectively barricade heat and smoke in fires and suggest using CFD software to study the design of air curtains in fire safety engineering. Hu et al. [21] used a numerical study to find that air curtains can effectively confine smoke and CO gases released from fires in the near-fire area. Gui et al. [22] found that air curtains affect wind fields, dust distribution, and particle number within about 15 m of an underground mining operation in the coal mine. These studies show that air curtains can provide a gas barrier effectively confine the flow of CO gases.

Analysis of the air curtain is crucial to finding the design parameters and suitable conditions for the air curtain to barrier the toxic and hazardous gases [23]. Wang et al. [24] found by CFD numerical simulation and experimental study that the highest sealing efficiency is obtained by installing a double-side air curtain device at the protective door of the refuge. Luo et al. [25] studied the smoke suppression effect of the opposed double-jet curtain in high-rise building fires and found that the smoke and CO released during the fire were confined in the corridor. Safarzadeh et al. [26] found that horizontal air curtains can effectively control the CO gas flow in building fires. Yu et al. [27] found that the width of the air curtain nozzle has a limited influence on the sealing effect. When the angle of the air curtain spraying to the fire source is 30°, the air curtain has the best effect in blocking smoke. Chen et al. [28] found that the effectiveness of preventing smoke is stronger the higher the jet velocity and the wider the air curtain. The effectiveness of controlling smoke gradually decreases as the jet angle increases. Razeghi et al. [29] studied the efficiency of the emergency ventilation system and discovered that by altering air curtain parameters and the emergency ventilation system, the diffusion of heat, CO, and $CO_2$ could be controlled for various fire sites and heat release rates. Zhang et al. [30] used CFD simulation software to study the variation of CO concentration, visibility, and temperature at different air curtain jet velocities in the same fire environment and found that when the jet velocity of the air curtain is 14 m/s, it is the most effective in restricting the smoke flow. It can be seen from the above analysis that the nozzle width, speed, installation position, and airflow angle of the air curtain have a significant influence on the blocking efficiency of the air curtain. There are two kinds of air curtains commonly used in mine refuge chambers, one is an air knife air curtain, and the other is a pipeline air curtain. Due to the lack of a unified standard for door wall construction and air curtain systems, the air curtain system varies greatly from refuge to refuge, some of which are shown in Figure 1.

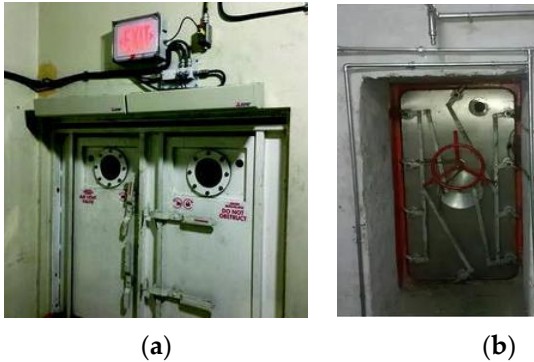

(**a**)                    (**b**)

**Figure 1.** Diagram of the structure and installation position of the mine refuge air curtain [31]. (**a**) Air knife air curtain installed on top side. (**b**) Pipeline air curtain installed on three sides.

At present, the research on pipeline air curtains in the mine refuge chamber is not mature and perfect. Jin et al. [32] found that when the refuge chamber door frame is 1.55 m wide and 1.00 m high, the optimal hole diameter and hole spacing of the pipeline air curtain is 1.2 and 40 mm, respectively. Xiao et al. [33] tested the performance of a pipeline air curtain system in refuge chambers in terms of CO gas barrier performance, showing a

barrier efficiency of 65% within 2.25 min when the air curtain was installed on both sides of the door frame and at a wind speed of 10.2 m/s. Zhang et al. [34] studied the influence of structure size, installation position, and airflow angle on the barrier effect. The results show that when the $CO_2$ concentration in the outside environment is 2%, the air curtain with a hole diameter of 1 mm and a hole spacing of 15 mm can block $CO_2$ gases in the tunnel by 55–60% after 5 min. It can be seen from the above that their research object is mainly $CO_2$ gases. It can be seen that the parameters to achieve the best barrier effect of the air curtain are different for different application scenarios. At the same time, they did not deeply study the relationship between structural parameters, velocity, and barrier effect.

The above study demonstrates that different application contexts and barrier item types require varying air curtain structural parameters, airflow velocity, installation positions, and airflow angles to obtain the greatest barrier effect. We are unable to immediately transfer air curtain systems from other application situations or conditions to the mine refuge, for this reason. To obtain the mathematical relationship between the parameter change and the efficiency of the air curtain, which further provides reference suggestions for pipeline structural design, this paper methodically investigates the influence of the aforementioned parameters on the efficiency of the air curtain.

## 2. Model Development

A 50-person refuge room is used as a physical model, and necessary modifications are made to the dimensions and boundary conditions of the model.

### 2.1. Description of Air Curtain System Model

Figure 2 illustrates that the mine refuge chamber includes a transition room and survival room, with the blast door of the transition room being closed during regular mine operations. In the event of a gas explosion or fire in the underground tunnel, evacuees reach the transition room door firstly from the outside environment, then open the blast door into the transition room, and then get into the survival room. Installed air curtains between the outside environment and the transition room can produce an airflow barrier to CO gas from entering the survival room [35–37].

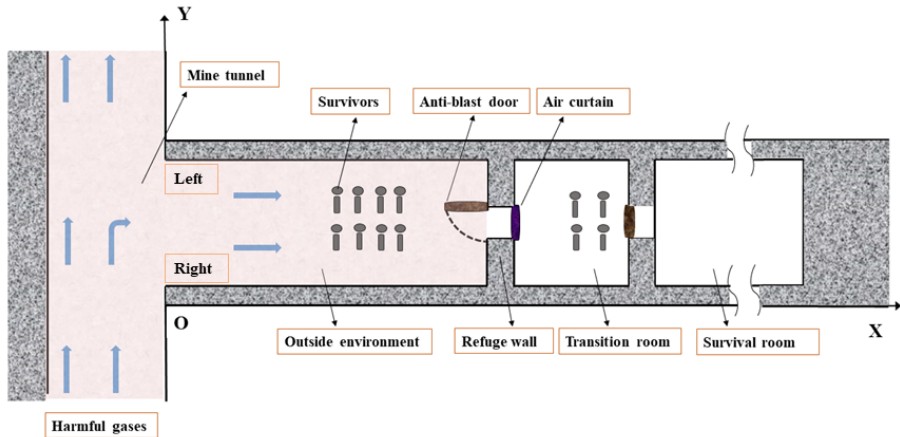

**Figure 2.** Refuge space division.

Figure 3 illustrates that the dimensions of the outside environment in XYZ-direction are respectively $12 \times 3.6 \times 3$ m, the dimensions of the transition room in XYZ-direction are, respectively, $4 \times 3.6 \times 3$ m, the dimensions of the door frame wall of the refuge room in XYZ-direction are respectively $0.8 \times 1.6 \times 0.72$ m. The transition room model used here is the same as in the previous publication, namely Ref. [34]. To more accurately study the air curtain barrier performance on CO gas, the size of the outside environment is modified in this paper, making the volume of the outside environment three times larger than the volume of the transition room, which ensures the accuracy of the simulation results [38,39].

The CO concentration above the transition room is higher than that below because the mass fraction of CO is smaller than air. The measurement point is arranged at 1# (14.72, 1.805, 2 m) to ensure the accuracy of the concentration of the measurement points in the transition room. Figure 4 illustrates that the air curtain is a row of round holes in a circular duct as outlets for airflow. The pipeline air curtain is installed close to the refuge door wall. Therefore, the air outlets on the air curtain pipeline are directly simplified as the air outlets on the wall, and the size of the hole diameter and hole distance is determined by the simulated working conditions.

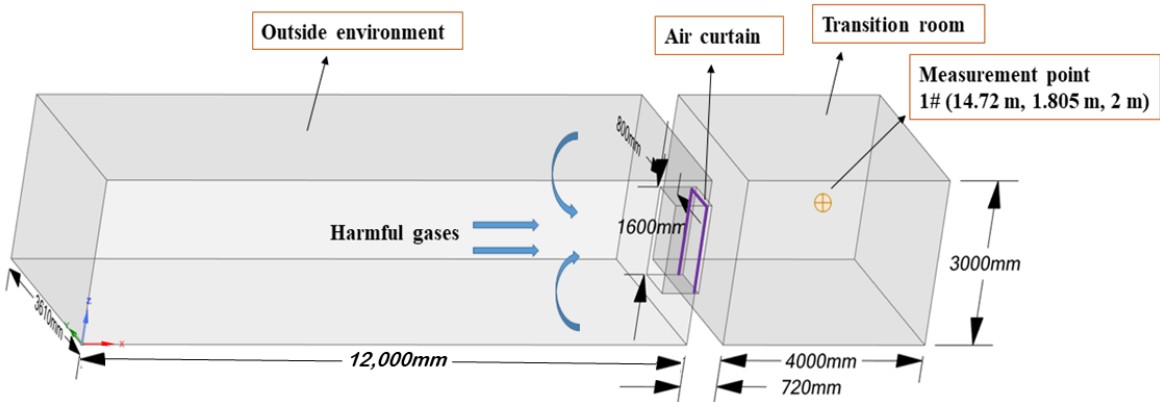

**Figure 3.** Geometric model of the mine refuge chamber.

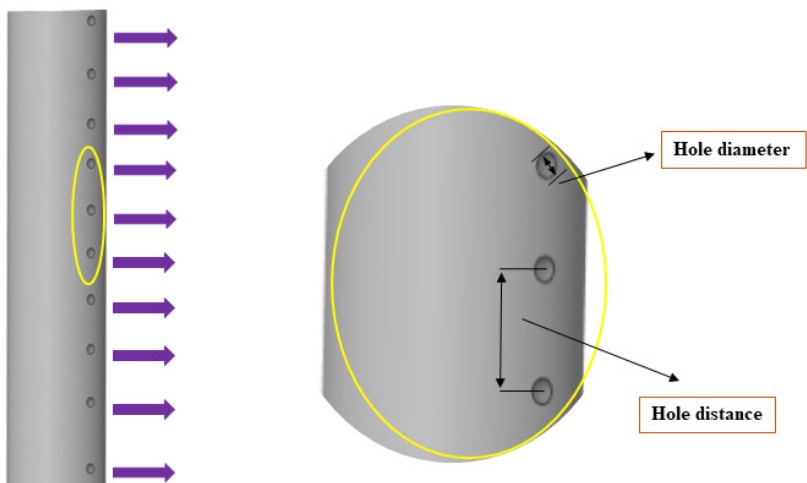

**Figure 4.** Model of air curtain.

## 2.2. Airflow Analysis at the Transition Room Entrance

Under the action of temperature difference, the air density inside and outside the transition room is different, which will have a large thermal pressure difference at the transition room door, and there will be a neutral surface in the middle of the transition room door without considering the effect of wind pressure. Figure 5 illustrates that above the neutral surface, high-concentration CO gas flows into the transition room. Below the neutral surface, the air flows out of the transition room. As the temperature difference increases, the heat and mass exchange rate near the neutral surface will increase. As the temperature difference decreases, the heat and mass exchange rate near the neutralization surface decrease. The change curve of CO concentration with time can be well explained by analyzing the airflow in front of the transition chamber door.

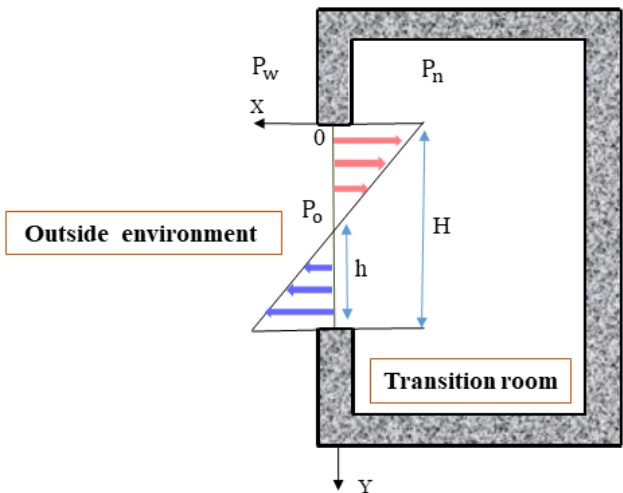

**Figure 5.** Hot pressure distribution diagram.

The coordinate system as shown in the figure is established. The expression for hot pressing:

$$p = gh(\rho_n - \rho_w) \tag{1}$$

In the above formula: $\rho_n$ is the air density inside the transition room, kg/m$^3$; $\rho_w$ is the air density of the outside environment, kg/m$^3$.

The neutralizing surface pressure is P$_0$, the pressure difference on the y-section:

$$\Delta p_1 = (h - x)(\rho_n - \rho_w)g \tag{2}$$

The flow rate due to differential pressure is:

$$\Delta p_1 = \frac{\rho \xi v^2}{2} \tag{3}$$

Combining Equations (3) and (4) yields:

$$V = \sqrt{\frac{2g(h - x)(\rho_n - \rho_w)}{\rho \xi}} \tag{4}$$

where $\xi$ is the resistance coefficient of the doorway; $V$ is the velocity in the x-direction at section y, m/s; $\rho$ is the density of air.

### 2.3. Mesh Dividing

Figure 6 illustrates that this paper uses the poly-hexcore method in fluent meshing to generate the body mesh, which divides the model into hybrid mesh composed of polyhedra and hexahedra. The poly-hexcore body mesh generation method enables the hexahedral mesh and the polyhedral mesh to achieve common node connection and supports the division of boundary layer meshes, thus further improving the overall quality of the mesh, effectively reducing the number of overall meshes and solving time. Using fluent meshing to mesh the computational model, the inlets of the air curtain are defined as facesize_1 by add local sizing, and the target mesh size is set to 0.4 mm. The whole computational model, including the outside environment and the transition room, is defined as facesize_2, and the target mesh size is set to 100 mm. In this way, the overall mesh quantity and the mesh quantity of the inlet of the air curtain can be controlled, and the inlets of the air curtain can be locally encrypted, thus improving the mesh quality. The non-orthogonality of the grid is 0.200267, the skewness of the grid is 0.21, and the maximum aspect ratio of the grid is 20.0213.

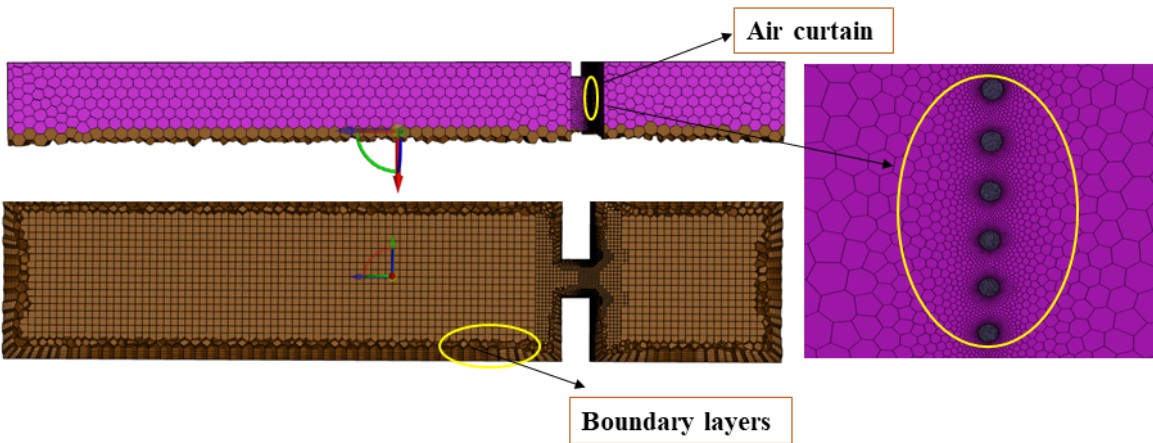

**Figure 6.** Grid display of the model.

Computational fluid dynamic simulations have strict mesh quality requirements. Grid quality directly affects the precision and accuracy of simulation results. Increasing the number of grids to improve grid quality also increases the time of simulation calculation. Hence the independence of the grids should be tested before the computational fluid dynamics simulations [40,41]. Six sets of air curtains with grids of 1,024,542, 1,617,526, 2,577,813, 3,141,872, 3,629,871, and 4014254 are installed on two sides of the refuge and tested for independence. Figure 7 illustrates that the CO concentration at the measurement point does not change when the number of grids exceeds 2,577,813, so the model with a grid number of 2,577,813 is used for the simulation.

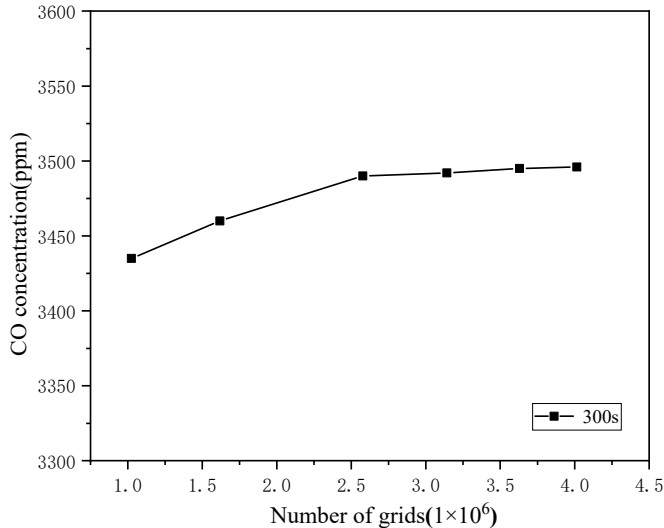

**Figure 7.** Grid independence test.

### 2.4. Boundary Conditions

In the event of a gas explosion or fire in the mine tunnel, the ventilation equipment at the workplace will be damaged to the extent, thus it is assumed that there is no wind flow at the working face. Meanwhile, the tunnel temperature rises, making the temperature of the outside environment higher than the transition room. The outside environmental temperature is set to 55 °C [42], and the transition room temperature is set to 25 °C [43]. The CO concentration of the outside environment studied in this paper is set at 10,000 ppm [44].

### 2.5. Solver Setup and the Governing Equation

The gas phase consists of air and carbon monoxide. the density of air varies with the temperature [45]. The air curtain inlets are velocity inlets, the velocity magnitude and angle are set with the working condition, and the initial temperature of the velocity outlet is set to 25 °C because the airflow is stored in the compression bottle of the transition room. The coupling of pressure and velocity is done by the SIMPLE algorithm, and the near-wall surface is set as a non-slip stationary wall surface. The viscous model is chosen from the SST $k-w$ model, where the turbulence effect is considered [46,47]. Continuity and momentum equations are used to analyze turbulent flows. CFD basic solution equation is as follows:

The continuous equation is as follows:

$$\frac{\partial \rho}{\partial t} + \nabla \cdot \rho \vec{u} = 0 \tag{5}$$

The momentum equation is as follows:

$$\rho\left(\frac{\partial \vec{u}}{\partial t} + \frac{1}{2}\nabla\left|\vec{u}\right|^2 - \vec{u} \times \omega\right) + \nabla p - \rho g = \vec{f} + \nabla\sigma \tag{6}$$

The energy equation is as follows:

$$\frac{\partial}{\partial t}(\rho h) + \nabla \cdot (\rho h \vec{u}) = \frac{\partial p}{\partial t} + \vec{u} \cdot \nabla p - \nabla \vec{q}_r + \nabla \cdot (k\nabla T) + \sum_i \nabla(h_i \rho D_i \nabla Y_i) \tag{7}$$

The component transport equation is as follows:

$$\frac{\partial}{\partial t}(\rho Y_i) + \nabla \cdot (\rho Y_i \vec{u}) = \nabla(\rho D_i \nabla Y_i) + \overset{\bullet}{m}_i^m \tag{8}$$

### 2.6. Validation of Computational Models

The CO2 concentration obtained using this numerical model is compared to the experimental measurement value of the concentration without the air curtain in the prior publication, namely Ref. [34], to demonstrate that the model is reliable and accurate. Zhang et al. [34] used a section of the outside environment in front of the transition room of the refuge to experimentally study the environment of a tunnel filled with toxic and hazardous gases. The outside environment is sealed with plastic film and the blast door between the transition room and the external environment is closed, at which point the outside environment is a confined space. Carbon dioxide was released into the outside environment through the dispersion pipe, and the fan is turned on for mixing. By detecting the $CO_2$ concentration in the outside environment, when the $CO_2$ concentration in the outside environment reaches 2% (20,000 ppm) uniformly, the release of $CO_2$ is stopped, and the fan is turned off. While opening the air curtain system, the explosion-proof door between the outside environment and the transition room is opened, and four carbon dioxide sensors are arranged in the outside environment to monitor the decreasing trend of $CO_2$ concentration, and three carbon dioxide sensors are arranged in the transition room to monitor the increasing trend of $CO_2$ concentration.

The present numerical model is found better in accordance with both qualitative and quantitative terms, with deviations varying in the range of 10 to 15%, which indicates it can be effectively used for the following research study, see Figure 8.

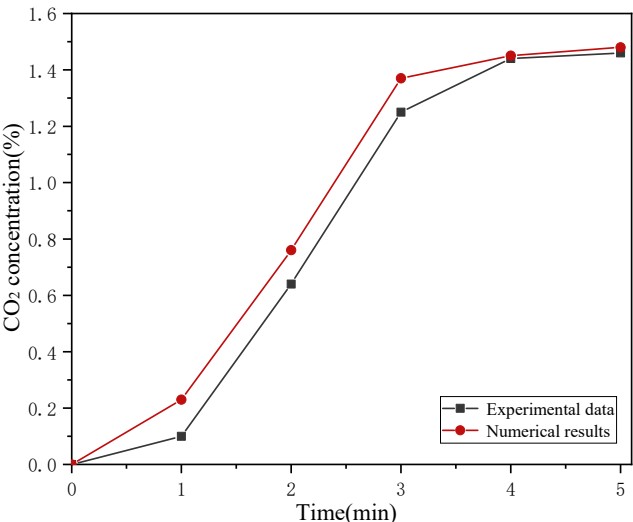

**Figure 8.** Comparison of the experimental and numerical results.

*2.7. Evaluation Metrics*

The barrier efficiency formula is as follows [34]:

$$\eta_z = \frac{Q_o - Q_m}{Q_o} \times 100\% \tag{9}$$

where $\eta_z$ represents the air curtain barrier efficiency (%), $Q_o$ represents the airflow through the door frame without air curtain (m$^3$/s), $Q_m$ represents the airflow through the air curtain door frame (m$^3$/s).

The airflow finally enters a certain volume of space $V$, so the formula is as follows:

$$\eta_z = \frac{(Q_o - Q_m)/V}{Q_o/V} \times 100\% \tag{10}$$

The formula is as follows when $V$ is a constant:

$$\eta_z = \frac{C_o - C_m}{C_o} \times 100\% \tag{11}$$

where $C_o$ represents the CO gases concentration in the transition room without an air curtain (ppm) and $C_m$ represents the CO gases concentration in the transition room with air curtain (ppm).

## 3. Simulation Results

Studying the effects of various installation positions, jet velocity, structural parameters, and angles on the performance of the air curtain CO barrier is based on the verified numerical model of the mine refuge chamber. This makes the results obtained from numerical simulations more convincing.

### 3.1. Effects of Installation Position and Jet Velocity

To investigate the effects of air curtains with various installation positions and jet velocities on CO gas barrier performance, we used the pipeline air curtain with a hole diameter of 6 mm and a hole spacing of 15 mm. Through analyzing the numerical simulation results we get the best installation position of the air curtain and the law of the impact of jet velocity on the air curtain efficiency.

### 3.1.1. Installed the Air Curtain on the Left Side

Figure 9 illustrates that installing air curtains can effectively barrier the CO gases getting into the transition room compared with no air curtain operation. The efficiency of the air curtain as a barrier for CO gas first rises and then falls as the jet velocity of the air curtain increases. The air curtain has the best barrier effect at a jet velocity of 18 m/s. At this time, the CO concentration in the transition room is 3100 ppm at 300 s.

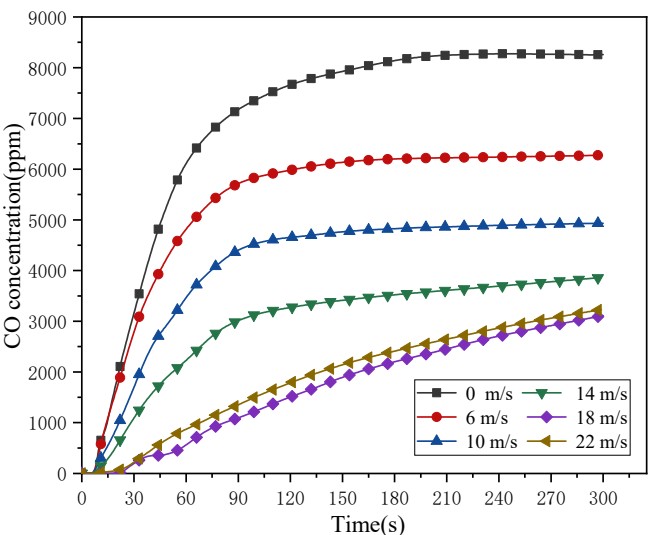

**Figure 9.** The CO concentration with air curtains installed on the left side.

There is no air curtain functioning since the air curtain jet velocity is 0 m/s. Under the effect of thermal pressure, the CO concentration in the transition room increased rapidly to 7175 ppm within 90 s. After 90 s, the CO concentration in the transition room increased slowly, CO concentration reaches 8300 ppm at 300 s. As the door of the refuge chamber opens, under the action of hot pressing, a strong heat and mass exchange takes place between the outside environment and the transition room. We can see that the CO concentration in the transition chamber rises rapidly at the beginning of the simulation. After a period of time, the decrease in temperature difference leads to the decrease of heat and mass exchange rate, and the rising rate of CO in the transition chamber decreases, as Figure 10 shows.

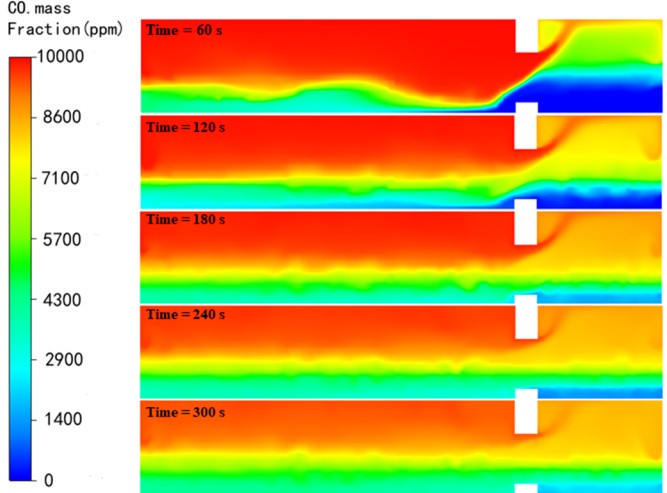

**Figure 10.** Cloud diagram of CO concentration in the transition room without air curtain at Y = 1.805 m.

Increase the velocity of the air curtain jet from 6 to 16 m/s, the CO concentration curve slope gradually becomes smaller in the period of 0–90 s, and the efficiency of blocking CO gases gradually increases. A buoyant jet known as an air curtain blends the effects of forced and natural convection. Consequently, the air curtain requires to have enough kinetic energy to maintain the inside and outside the neutral plane pressure differences. The air curtain cannot form a complete airflow barrier from top to bottom owing to the low jet airflow energy. The jet airflow energy becomes higher due to the increased jet velocity, gradually forming a top-down airflow to barrier CO gases exchange. For this reason, the air curtain sealing effect becomes better. The air curtain jet velocity needs to be 18 m/s to have the best sealing effect, and the CO concentration in the transition room is 3100 ppm at 300 s. Currently, the CO concentration in the transition room varies as a straight line with time. The kinetic energy of air curtain flow increases with the increase of velocity. The high kinetic energy gas flow barrier makes the high concentration of CO gas under hot pressure enter the transition room at a specific rate. The efficiency of the air curtain to block CO gases decreases as the velocity increases to 22 m/s. Because the air jet velocity is too high, the kinetic energy of the airflow is too high, which makes the airflow hit the ground of the refuge chamber. Thus, CO gases accelerated to flow into the transition room. As a result, the air curtain barrier efficiency declines. The barrier performance of the air curtain is found to not be linearly related to the jet velocity based on the studies mentioned above.

### 3.1.2. Installed the Air Curtain on the Two Sides

Figure 11 shows that the blocking efficacy of the air curtains does not significantly increase with the increase in jet velocity when the jet velocity of the air curtains installed on both sides of the refuge chamber exceeds 14 m/s. And air curtain barrier performance approximately remains unchanged as well. Therefore, it is considered that the optimal jet velocity of the air curtain is 14 m/s.

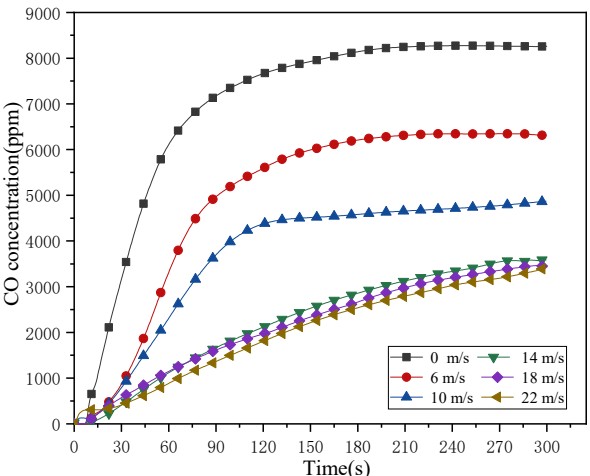

**Figure 11.** The CO concentration with air curtains installed on the two sides (left) and (right).

The air curtain CO barrier effect is positively correlated with the jet velocity once the jet velocity is less than 14 m/s. Air curtain effectiveness can be considerably enhanced by increasing the jet velocity. The number of airflow outlets installed with air curtains on two sides is twice the number of left-installed air curtains airflow. As a result, the air curtain jet velocity which is installed on two sides will be lower, under the same total kinetic energy of the airflow. The CO barrier operates most efficiently when the air curtain jet velocity installed on two sides is 14 m/s. The CO concentration in the transition room is 3600 ppm at 300 s. At this time, the change of CO concentration with time is roughly in a straight line. The performance of the air curtain barrier does not significantly improve as the jet velocity rises from 14 to 22 m/s. Because it does not match the actual project, we are unable

to constantly increase the jet velocity to study the effectiveness of the air curtain barrier. Therefore, the performance of air curtain barriers installed on both sides is lower than that installed on the left.

### 3.1.3. Installed the Air Curtain on the Top Side

Figure 12 illustrates that installed on the top side of the refuge door frame causes the barrier efficiency of the air curtain first increases and then decrease as jet velocity increases. The air curtain is most effective when the jet velocity is 22 m/s, and the CO concentration in the transition room is 2740 ppm at 300 s.

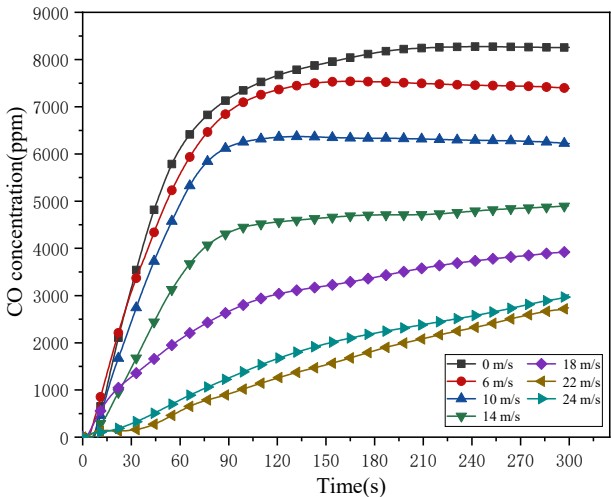

**Figure 12.** The CO concentration with air curtains installed on the top side.

Compared with the air curtain installed in other positions, the air curtain installed at the top side provides the best barrier efficiency because the air flows into the outside environment from the lower side of the neutral surface while the high concentration of CO gases in the outside environment flows into the transition room from the upper side of the neutral surface, driven by heated pressure. The air curtain erected on the top side of the refuge chamber has the best blocking effect because it can effectively stop CO gases from entering the transition chamber on the top side of the neutral plane. The ideal velocity of the air curtain is 22 m/s. The number of air curtain inlets installed on the top side is the lowest compared to those installed in other locations. The air curtain must increase the jet velocity to have more kinetic energy in order to resist the effect of external heat pressure and create the best barrier effect.

It is evident from the data above that the air curtain installation position and jet velocity greatly affect the barrier CO efficiency. Various air curtains with different jet velocities have different barrier effectivity at the same installation point. For the same jet velocity, the air curtain barrier efficacy varies depending on the installation position. The efficiency of the barrier cannot always be increased by increasing the air curtain jet velocity when the air curtain position is known. In all of the above conditions, the air curtain with 22 m/s jet velocities erected on the top side offers the best barrier effect.

### 3.2. Effect of Structural Parameters

Air curtains are installed on the top side of the refuge door, keeping the jet velocity constant at 22 m/s. Figures 13–15 illustrate the relationship between the CO concentration and the air curtain hole diameters of 4, 5, and 6 mm, and also their hole spacing of 10, 15, 20, 25, and 30 mm.

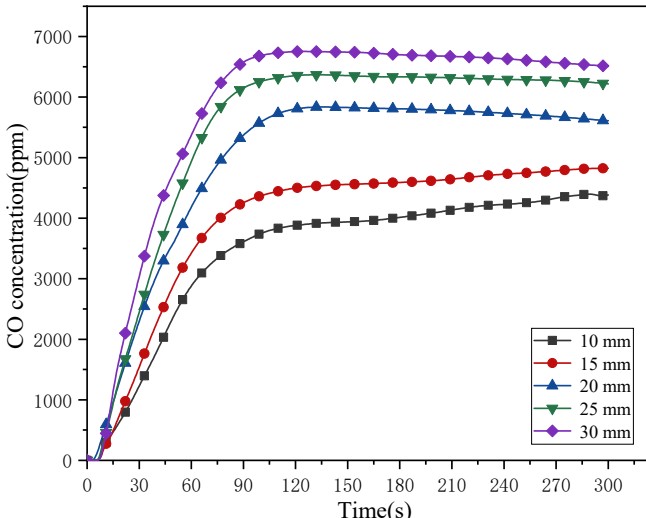

**Figure 13.** The CO concentration in transition room with 4 mm hole diameters.

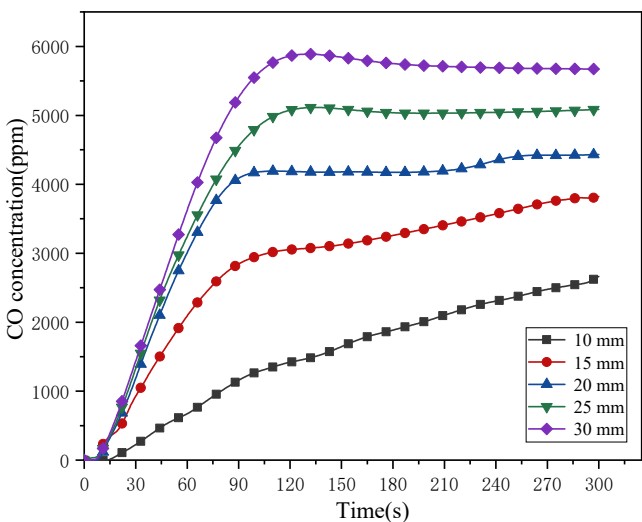

**Figure 14.** The CO concentration in transition room with 5 mm hole diameters.

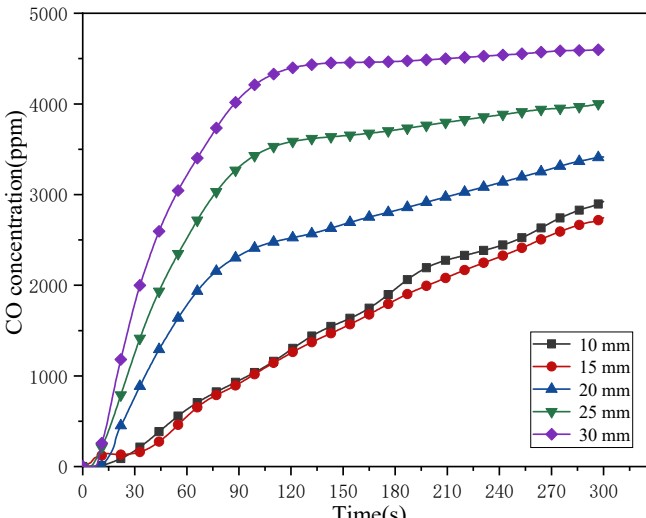

**Figure 15.** The CO concentration in transition room with 6 mm hole diameters.

Figures 13–15 demonstrate that when the air curtain hole spacing is increased while the hole diameter remains constant, the concentration of CO in the transition room rises over time. The 300 s transition room CO concentration exhibits a decreasing trend when the air curtain hole spacing remains constant and the air curtain hole diameter is increased. The total kinetic energy of the air curtain flow is proportional to the hole diameter when hole spacing, and jet velocity are both constant. In other words, the airflow barrier cannot form at the top and bottom because the total kinetic energy of the airflow ejected from the air curtain is lower for the smaller air curtain hole diameter. Thermal pressure causes the exchange of thermal mass between the outside environment and the transition room to aggravate, and this weakens the air curtain barrier effect. Increase the hole diameter and the total kinetic energy of the air curtain that is ejected will also rise. This will allow the air curtain to not only form a top-down airflow barrier but also to better withstand the effects of external thermal pressure as well, gradually increasing its barrier effect. The relationship between the ejected airflow kinetic energy and hole spacing, however, is inversely proportional when the air curtain hole diameter and jet velocity all are known. If the air curtain hole spacing is decreased, the total kinetic energy of the air curtain sprayed out of the curtain increases while its barrier effectiveness increases. Conversely, if the air curtain hole spacing is increased, the total kinetic energy of the air curtain sprayed out of the curtain decreases while its barrier effectiveness decreases.

The red sphere represents the three-dimensional coordinates of hole diameter, hole spacing, and air curtain barrier efficiency. The mathematical relationship between air curtain efficiency and the three holes with diameters of 4, 5, and 6 mm and the five holes with spacings of 10, 15, 20, 25, and 30 mm is shown in Figure 16.

$$Z = 50.45 - 1.416x + 11.92y \tag{12}$$

In this formula, $Z$ represents the barrier efficiency of air curtains, $x$ represents the spacing between air curtain holes, and y for the diameter of the air curtain. It is discovered that the efficiency of the air curtain increases with increasing hole diameter and decreases with increasing air curtain hole spacing. The formula provides a reference for the optimal design of the air curtain pipeline structure.

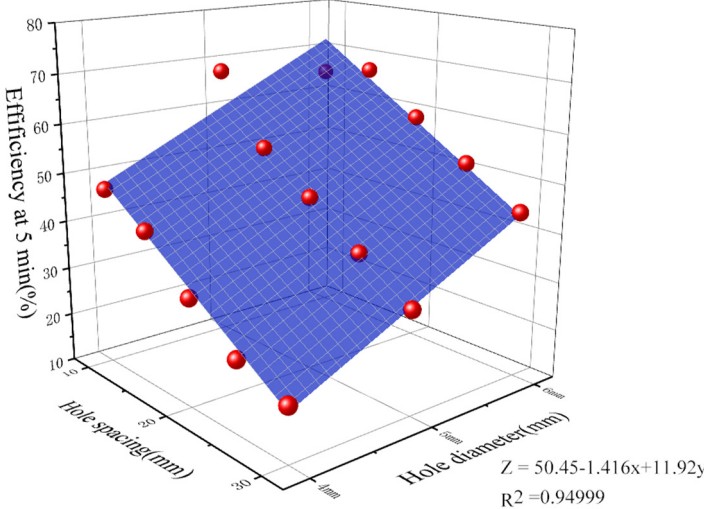

**Figure 16.** A mathematical relationship between hole diameter, hole spacing, and air curtain efficiency.

### 3.3. Effect of Jet Angles

Keeping the hole diameter of 6 mm, the hole spacing of 15 mm, and the jet velocity of 22 m/s as constants. And the air curtain is located on the top side of the chamber door frame. Figure 17 shows that jet angle affects the barrier performance of the air curtain. When the air curtain jet angle is 10°, that is, the air curtain into the outside environment at

an angle of 10°, and 300 s transition room CO concentration of 2630 ppm, the air curtain at this time has the best barrier effect.

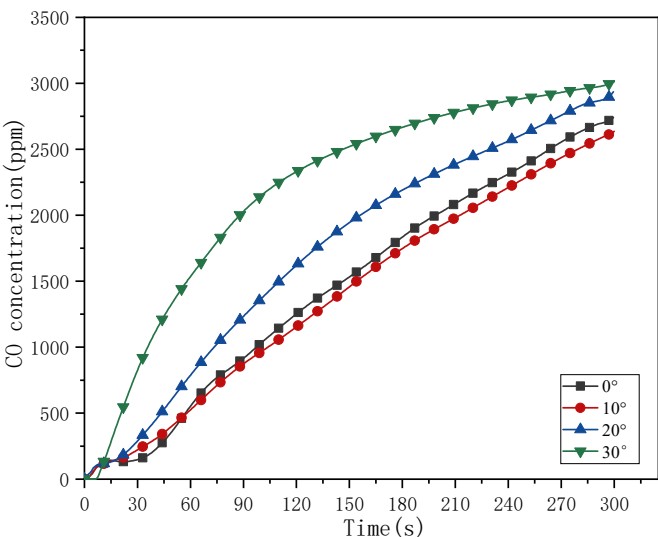

**Figure 17.** The CO concentration in transition room at different jet angles.

When the air curtain jet angle is 0°, the air curtain airflow flows vertically to the chamber door ground floor, which results in a lower concentration than when the air curtain jet angle is 10°. The airflow from the air curtain will tilt toward the transition room because of the buoyancy created by hot pressing, partially reducing the air curtain barrier effect. As a result of heat pressing, the air curtain airflow blows vertically to the ground when the jet angle it has towards high concentration CO is 10°. The air curtain has an excellent blocking effect, and the jet angle is 10°. Figure 18 shows the velocity cloud diagram of airflow in the air curtain at a jet angle of 10°. The air curtain is more effective at obstructing CO because, as shown in the picture, it blows to the ground vertically while being affected by the thermal pressure of the environment outside. Figure 19 shows the CO concentration cloud diagram at varying moments in the transition room now that the air curtain jet angle is 10°. The cloud diagram shows that installing an air curtain to block CO has a significant impact as compared to not performing it. The length to be sealed also increases as the angle of the curtain jet increases since it increases the distance from the nozzle to the refuge ground. To create a complete air curtain, it is necessary to raise the jet velocity of the air curtain and the overall kinetic energy of the airflow. The CO barrier effect decreases if the jet velocity stays constant and the air curtain jet angle keeps rising.

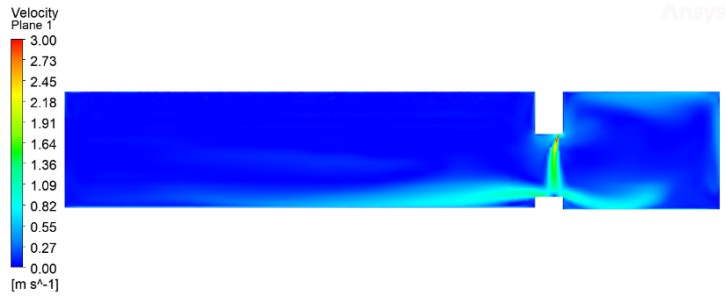

**Figure 18.** 300 s air curtain velocity cloud diagram when the jet angle is 10° at Y = 1.805 m.

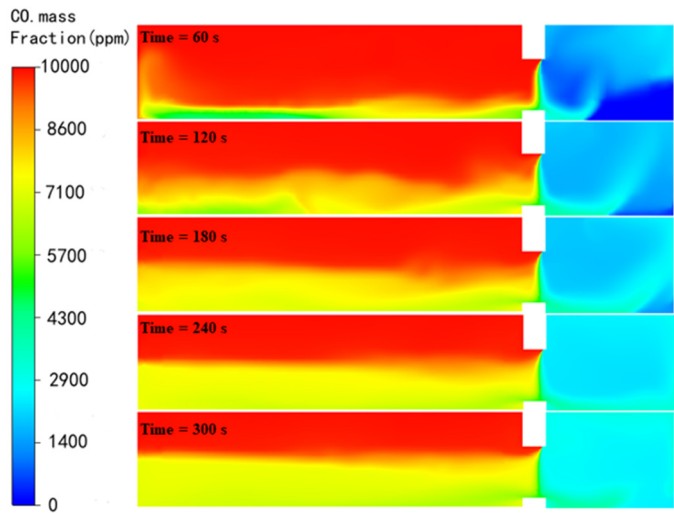

**Figure 19.** The CO concentration cloud diagram of the transition room when the jet angle is 10° at Y = 1.805 m.

By analyzing each factor affecting the performance of the air curtain in blocking CO gas, we can not only draw a regular relationship between each influencing factor and the blocking efficiency. A specific conclusion can also be drawn. The best CO barrier effect is achieved by installing an air curtain at the top with a jet velocity of 22 m/s and a jet angle of 10°.

## 4. Discussion

Thirty-three different conditions can be used to calculate the air curtain system barrier effectiveness at 5 min. The barrier effect of the air curtain is significantly influenced by the installation site and jet velocity, as shown in Table 1. The influence of the air curtain jet angle on the effectiveness of the air curtain barrier is not immediately apparent. The barrier effect of the air curtain installation position on the top side of the refuge door frame was 68.1% at a jet velocity of 22 m/s and a jet angle of 10° in an outside environment with a CO concentration of 1% for a hole diameter of 6 mm and a hole spacing of 15 mm.

**Table 1.** The barrier efficiency of pipeline air curtain at 5 min in 33 kinds of conditions.

| Condition | Installation Location | Airflow Velocity (m/s) | Hole Diameter (mm) | Hole Spacing (mm) | Airflow Angle | Efficiency at 5 min |
|---|---|---|---|---|---|---|
| 1 | Left side | 6 | 6 | 15 | 0° | 24.0% |
| 2 | Left side | 10 | 6 | 15 | 0° | 40.3% |
| 3 | Left side | 14 | 6 | 15 | 0° | 53.2% |
| 4 | Left side | 18 | 6 | 15 | 0° | 62.4% |
| 5 | Left side | 22 | 6 | 15 | 0° | 60.9% |
| 6 | Two sides | 6 | 6 | 15 | 0° | 23.6% |
| 7 | Two sides | 10 | 6 | 15 | 0° | 41.0% |
| 8 | Two sides | 14 | 6 | 15 | 0° | 57.7% |
| 9 | Two sides | 18 | 6 | 15 | 0° | 58.2% |
| 10 | Two sides | 22 | 6 | 15 | 0° | 58.6% |
| 11 | Top side | 6 | 6 | 15 | 0° | 10.5% |
| 12 | Top side | 10 | 6 | 15 | 0° | 24.7% |
| 13 | Top side | 14 | 6 | 15 | 0° | 40.7% |
| 14 | Top side | 18 | 6 | 15 | 0° | 52.5% |
| 15 | Top side | 22 | 6 | 15 | 0° | 66.8% |

**Table 1.** *Cont.*

| Condition | Installation Location | Airflow Velocity (m/s) | Hole Diameter (mm) | Hole Spacing (mm) | Airflow Angle | Efficiency at 5 min |
|---|---|---|---|---|---|---|
| 16 | Top side | 24 | 6 | 15 | 0° | 63.4% |
| 17 | Top side | 22 | 4 | 10 | 0° | 46.9% |
| 18 | Top side | 22 | 4 | 15 | 0° | 41.6% |
| 19 | Top side | 22 | 4 | 20 | 0° | 32.1% |
| 20 | Top side | 22 | 4 | 25 | 0° | 24.7% |
| 21 | Top side | 22 | 4 | 30 | 0° | 21.7% |
| 22 | Top side | 22 | 5 | 10 | 0° | 67.7% |
| 23 | Top side | 22 | 5 | 15 | 0° | 53.7% |
| 24 | Top side | 22 | 5 | 20 | 0° | 46.3% |
| 25 | Top side | 22 | 5 | 25 | 0° | 38.4% |
| 26 | Top side | 22 | 5 | 30 | 0° | 31.3% |
| 27 | Top side | 22 | 6 | 10 | 0° | 64.6% |
| 28 | Top side | 22 | 6 | 20 | 0° | 58.6% |
| 29 | Top side | 22 | 6 | 25 | 0° | 51.4% |
| 30 | Top side | 22 | 6 | 30 | 0° | 44.3% |
| 31 | Top side | 22 | 6 | 15 | 10° | 68.1% |
| 32 | Top side | 22 | 6 | 15 | 20° | 64.2% |
| 33 | Top side | 22 | 6 | 15 | 30° | 63.5% |

Currently, people install air curtains in front of refuge chambers directly. This way cannot maximize the barrier effectiveness. At present, the hole diameter of the air curtain system used in mine is very small. In a dusty mine airflow environment, the airflow ports of the air curtain are easily blocked, thus reducing the effectiveness of the air curtain barrier. Compared to the air curtains currently installed in refuge chambers, the large hole diameter pipeline air curtain studied in this paper is not only more efficient in blocking CO, but also more stable in performance. The study of the impact of the installation position, angle, and jet velocity on CO barrier effectiveness serves as a guide for the installation and commissioning of air curtains. It provides guidance on the design of the structural parameters of the air curtain by studying how the structural factors of the air curtain affect the blocking effect.

## 5. Conclusions

This study analyzes the CO barrier effectiveness of the air curtain in the mine refuge chamber with varying installation positions, jet velocity, angle, and structural parameters. The specific findings that can be drawn from numerical analysis are as follows:

1. Increasing the jet velocity of the air curtain does not always increase the barrier effect of the air curtain; increasing the jet angle of the air curtain, the barrier effect of the air curtain first increases and then decreases;
2. A 6 mm hole diameter air curtain with a 15 mm hole spacing was installed on the top side of the refuge door, with a jet velocity of 22 m/s and a jet angle of 10°, giving the best CO barrier;
3. The correlation equation of predicted air curtain hole diameter and hole distance with air curtain efficiency is proposed to provide guidance for the structural design of air curtains in other applications.

**Author Contributions:** All the authors made contributions to the concept and design of the article. methodology, Z.Z. and Z.S.; software, Z.S.; formal analysis Z.Z.; data curation, J.L.; writing—original draft preparation, Z.S.; writing—review and editing, Z.S.; visualization, R.M.; supervision, H.M.; Project administration, X.T.; funding acquisition, Z.Z. All authors have read and agreed to the published version of the manuscript.

**Funding:** The author would like to thank the financial support the National Natural Science Foundation of China (No. 52168013).

**Informed Consent Statement:** Informed consent was obtained from all subjects involved in the study.

**Data Availability Statement:** Data available in a publicly accessible repository.

**Conflicts of Interest:** The authors declare no conflict of interest. The funders had no role in the design of the study; in the collection, analyses, or interpretation of data; in the writing of the manuscript; or in the decision to publish the results.

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
