# Peer review of "Numerical Simulation of Co-Barrier Efficiency of Air Curtains in Mine Refuge Chambers"

_applsci, doi:10.3390/app13020993_

Round 1

Reviewer 1 Report

The authors need to significantly improve the manuscript before it can be recommended for publications. Specific points are:

1. The authors must clearly and convincingly articulate the novelty of this work in the Introduction

2. The caption of Fig. 5 is just one word. This is inappropriate. 

3. There are instances where paragraphs are formed from a single sentence. This is also inappropriate. It makes the readability of the manuscript quite poor.

4. The word "experimental" appears in the caption and the legends of Figure 7. However, the methods/materials used to conduct experiments and collect the experimental data presented in Figure 7 are not mentioned in the manuscript.     

Reviewer 2 Report

The present paper presents CFD study of the use of air-curtains in mine refuse chambers in case of mine blasts. The paper has used CO gas as the harmful gas which can flow inside the refuge chambers. The study performed is significant and holds a great interest in the practical scenario. But before the paper can be accepted, the authors need to address a few minor comments and incorporate those into the paper. Below are my comments:

1. The authors should mention the mesh quality value in terms of non-orthogonality, skewness and aspect ratio.

2. Page 6, line 155 is unclear: "The overall 155 size of the model is set to the maximum size of 100 mm" which model and what does size 155 mean?

3. The authors have performed a mesh independence study, but what is the basis of the increment of mesh numbers?

4. Check line 174; what does 2010 mean?

5. Authors have used Boussinesq approximation to solve for buoyancy force, but what is the condition up to which this is valid? Anyways the authors have considered CO and air as an ideal gas, then why not solve it in full compressible mode?

6. As the entire study focuses on the jet velocity and angle of the velocity at the hole, it becomes important to resolve the flow physics near that location. What is the mesh sizing near the holes in air-curtains? Will it be sufficient enough to capture the flow physics?

7. What is the basis for choosing hole diameter and hole spacing in air barriers? What will happen if the hole diameter is further increased?

8. Whether the BC for holes is velocity inlet or velocity outlet.

9. The authors sometimes use terminology like hole or nozzle for velocity inlet in the air-curtain; please follow any one terminology in the entire manuscript. 

10. The authors should also explain which regression analysis is conducted to get the correlation. 

Round 2

Reviewer 1 Report

Accept

Reviewer 2 Report

The authors have answered all the queries raised.